# Monte Carlo Tree Search based Variable Selection for High Dimensional Bayesian Optimization

**Lei Song**,* **Ke Xue**,* **Xiaobin Huang, Chao Qian**†
State Key Laboratory for Novel Software Technology,
Nanjing University, Nanjing 210023, China
{songl, xuek, huangxb, qianc}@lamda.nju.edu.cn

## Abstract

Bayesian optimization (BO) is a class of popular methods for expensive black-box optimization, and has been widely applied to many scenarios. However, BO suffers from the curse of dimensionality, and scaling it to high-dimensional problems is still a challenge. In this paper, we propose a variable selection method MCTS-VS based on Monte Carlo tree search (MCTS), to iteratively select and optimize a subset of variables. That is, MCTS-VS constructs a low-dimensional subspace via MCTS and optimizes in the subspace with any BO algorithm. We give a theoretical analysis of the general variable selection method to reveal how it can work. Experiments on high-dimensional synthetic functions and real-world problems (i.e., NAS-bench problems and MuJoCo locomotion tasks) show that MCTS-VS equipped with a proper BO optimizer can achieve state-of-the-art performance.

## 1 Introduction

In many real-world tasks such as neural architecture search (NAS) [41] and policy search in reinforcement learning (RL) [6], one often needs to solve the expensive black-box optimization problems. Bayesian optimization (BO) [2, 11, 23, 32] is a sample-efficient algorithm for solving such problems. It iteratively fits a surrogate model, typically Gaussian process (GP), and maximizes an acquisition function to obtain the next point to evaluate. While BO has been employed in a wide variety of settings, successful applications are often limited to low-dimensional problems.

Recently, scaling BO to high-dimensional problems has received a lot of interest. Decomposition-based methods [13, 15, 17, 26, 31] assume that the high-dimensional function to be optimized has a certain structure, typically the additive structure. By decomposing the original high-dimensional function into the sum of several low-dimensional functions, they optimize each low-dimensional function to obtain the point in the high-dimensional space. However, it is not easy to decide whether a decomposition exists as well as to learn the decomposition.

Other methods often assume that the original high-dimensional function with dimension $D$ has a low-dimensional subspace with dimension $d \ll D$, and then perform the optimization in the low-dimensional subspace and project the low-dimensional point back for evaluation. For example, embedding-based methods [20, 27, 42] use a random matrix to embed the original space into the low-dimensional subspace. Another way is to select a subset of variables directly, which can even avoid the time-consuming matrix operations of embedding-based methods. For example, Dropout [21] selects $d$ variables randomly in each iteration. Note that for both embedding and variable selection methods, the parameter $d$ can have a large influence on the performance, which is, however, difficult to set in real-world problems.

---

*Equal Contribution
†Corresponding Author

36th Conference on Neural Information Processing Systems (NeurIPS 2022).

In this paper, we propose a new Variable Selection method using Monte Carlo Tree Search (MCTS), called MCTS-VS. MCTS is employed to partition the variables into important and unimportant ones, and only those selected important variables are optimized via any black-box optimization algorithm, e.g., vanilla BO [32] or TuRBO [10]. The values of unimportant variables are sampled using historical information. Compared with Dropout-BO, MCTS-VS can select important variables automatically.

We also provide regret and computational complexity analyses of general variable selection methods, showing that variable selection can reduce the computational complexity while increasing the cumulative regret. Our regret bound generalizes that of GP-UCB [38] which always selects all variables, as well as that of Dropout [21] which selects $d$ variables randomly in each iteration. The results suggest that a good variable selection method should select as important variables as possible.

Experiments on high-dimensional synthetic functions and real-world problems (i.e., NAS and RL problems) show that MCTS-VS is better than the previous variable selection method Dropout [21], and can also achieve the competitive performance to state-of-the-art BO algorithms. Furthermore, its running time is small due to the advantage of variable selection. We also observe that MCTS-VS can select important variables, explaining its good performance based on our theoretical analysis.

## 2 Background

### 2.1 Bayesian Optimization

We consider the problem $\max_{\boldsymbol{x} \in \mathcal{X}} f(\boldsymbol{x})$, where $f$ is a black-box function and $\mathcal{X} \subseteq \mathbb{R}^D$ is the domain. The basic framework of BO contains two critical components: a surrogate model and an acquisition function. GP is the most popular surrogate model. Given the sampled data points $\{(\boldsymbol{x}^i, y^i)\}_{i=1}^{t-1}$, where $y^i = f(\boldsymbol{x}^i) + \epsilon^i$ and $\epsilon^i \sim \mathcal{N}(0, \eta^2)$ is the observation noise, GP at iteration $t$ seeks to infer $f \sim \mathcal{GP}(\mu(\cdot), k(\cdot, \cdot) + \eta^2 \mathbf{I})$, specified by the mean $\mu(\cdot)$ and covariance kernel $k(\cdot, \cdot)$, where $\mathbf{I}$ is the identity matrix of size $D$. After that, an acquisition function, e.g., Probability of Improvement (PI) [19], Expected Improvement (EI) [16] or Upper Confidence Bound (UCB) [38], is used to determine the next query point $\boldsymbol{x}^t$ while balancing exploitation and exploration.

### 2.2 High-dimensional Bayesian Optimization

Scaling BO to high-dimensional problems is a challenge due to the curse of dimensionality and the computation cost. As the dimension increases, the search space increases exponentially, requiring more samples, and thus more expensive evaluations, to find a good solution. Furthermore, the computation cost of updating the GP model and optimizing the acquisition function will be very time-consuming [30]. There have been a few common approaches to tackle high-dimensional BO with different assumptions.

**Decomposition.** Assuming that the function can be decomposed into the sum of low-dimensional functions with disjoint subspaces, Kandasamy et al. [17] proposed the Add-GP-UCB algorithm to optimize those low-dimensional functions separately, which was further generalized to overlapping subspaces [26, 31]. Wang et al. [43] proposed ensemble BO that uses an ensemble of additive GP models for scalability. Han et al. [13] constrained the dependency graphs of decomposition to tree structures to facilitate the decomposition learning and optimization. For most problems, however, the decomposition is unknown, and also difficult to learn.

**Embedding.** Assuming that only a few dimensions affect the high-dimensional function significantly, embedding-based methods embed the high-dimensional space into a low-dimensional subspace, and optimize in the subspace while projecting the point back for evaluation. REMBO and its variants use a random matrix to embed the search space into a low-dimensional subspace [3, 4, 42]. Nayebi et al. [27] used a hash-based method for embedding. Letham et al. [20] proposed ALEBO, focusing on several misconceptions in REMBO to improve the performance. The VAE-based approaches were also employed to project a structured input space (e.g., graphs and images) to a low-dimensional subspace [12, 22].

**Variable Selection.** Based on the same assumption as embedding, variable selection methods iteratively select a subset of variables to build a low-dimensional subspace and optimize through BO. The selected variables can be viewed as important variables that are valuable for exploitation, or having high uncertainty that are valuable for exploration. A classical method is Dropout [21],

which randomly chooses $d$ variables in each iteration. Spagnol et al. [37] uses Hilbert Schmidt Independence criterion to guide variable selection. When evaluating the sampled point, the values of those unselected variables are obtained by random sampling or using historical information. VS-BO [33] selects variables with larger estimated gradients and uses CMA-ES [14] to obtain the values of unselected variables. Note that variable selection can be faster than embedding, because the embedding cost (e.g., matrix inversion) is time-consuming for high-dimensional optimization.

Both embedding and variable selection methods need to specify the parameter $d$, i.e., the dimension of low-dimensional subspace, which will affect the performance significantly, but is not easy to set. There are also some methods to improve the basic components of BO directly for high-dimensional problems. For example, DNGO [36] uses the neural network as an alternative of GP to speed up inference; BO-PP [29] generates pseudo-points (i.e., data points whose objective values are not evaluated) to improve the GP model; SAASBO [9] uses sparsity-inducing prior to perform variable selection implicitly, making the coefficients of unimportant variables near to zero and thus restraining over-exploration on these variables. Note that different from Dropout and our proposed MCTS-VS, SAASBO still optimizes all variables, and also due to its high computational cost of inference, it is very time-consuming as reported in [9]. These methods can be combined with the above-mentioned dimensionality reduction methods, which may bring further improvement.

### 2.3 Monte Carlo Tree Search

MCTS [5] is a tree search algorithm based on random sampling, and has shown great success in high-dimensional tasks, such as Go [34, 35]. A tree node represents a state, describing the current situation, e.g., the position in path planning. Each tree node $X$ stores a value $v_X$ representing its goodness, and the number $n_X$ that it has been visited. They are used to calculate UCB [1], i.e.,

$$v_X + 2C_p\sqrt{2(\log n_p)/n_X}, \tag{1}$$

where $C_p$ is a hyper-parameter, and $n_p$ is the number of visits of the parent of $X$. UCB considers both exploitation and exploration, and will be used for node selection.

MCTS iteratively selects a leaf node of the tree for expansion. Each iteration can be divided into four steps: *selection*, *expansion*, *simulation* and *back-propagation*. Starting from the root node, selection is to recursively select a node with larger UCB until a leaf node, denoted as $X$. Expansion is to execute a certain action in the state represented by $X$ and transfer to the next state, e.g., move forward and arrive at a new position in path planning. We use the child node $Y$ of $X$ to represent the next state. Simulation is to obtain the value $v_Y$ via random sampling. Back-propagation is to update the value and the number of visits of $Y$'s ancestors.

To tackle high-dimensional optimization, Wang et al. [40] proposed LA-MCTS, which applies MCTS to iteratively partition the search space into small sub-regions, and optimizes only in the good sub-regions. That is, the root of the tree represents the entire search space $\Omega$, and each tree node $X$ represents a sub-region $\Omega_X$. The value $v_X$ is measured by the average objective value of the sampled points in the sub-region $\Omega_X$. In each iteration, after selecting a leaf node $X$, LA-MCTS performs the optimization in $\Omega_X$ by vanilla BO [32] or TuRBO [10], and the sampled points are used for clustering and classification to bifurcate $\Omega_X$ into two disjoint sub-regions, which are "good" and "bad", respectively. Note that the sub-regions are generated by dividing the range of variables, and their dimensionality does not decrease, which is still the number of all variables. Wang et al. [40] have empirically shown the good performance of LA-MCTS. However, as the dimension increases, the search space increases exponentially, and more partitions and evaluations are required to find a good solution, making the application of LA-MCTS to high-dimensional optimization still limited.

## 3 MCTS-VS Method

In this section, we propose a Variable Selection method based on MCTS for high-dimensional BO, briefly called MCTS-VS. The main idea is to apply MCTS to iteratively partition all variables into important and unimportant ones, and perform BO only for those important variables. Let $[D] = \{1, 2, \ldots, D\}$ denote the indexes of all variables $\boldsymbol{x}$, and $\boldsymbol{x}_{\mathbb{M}}$ denote the subset of variables indexed by $\mathbb{M} \subseteq [D]$.

We first introduce a $D$-dimensional vector named *variable score*, which is a key component of MCTS-VS. Its $i$-th element represents the importance of the $i$-th variable $x_i$. During the running

process of MCTS-VS, after optimizing a subset $\boldsymbol{x}_{\mathbb{M}}$ of variables where $\mathbb{M} \subseteq [D]$ denotes the indexes of the variables, a set $\mathcal{D}$ of sampled points will be generated, and the pair $(\mathbb{M}, \mathcal{D})$ will be recorded into a set $\mathbb{D}$, called *information set*. The variable score vector is based on $\mathbb{D}$, and calculated as

$$\boldsymbol{s} = \left( \sum_{(\mathbb{M},\mathcal{D}) \in \mathbb{D}} \sum_{(\boldsymbol{x}^i, y^i) \in \mathcal{D}} y^i \cdot g(\mathbb{M}) \right) \Big/ \left( \sum_{(\mathbb{M},\mathcal{D}) \in \mathbb{D}} |\mathcal{D}| \cdot g(\mathbb{M}) \right), \qquad (2)$$

where the function $g : 2^{[D]} \rightarrow \{0,1\}^D$ gives the Boolean vector representation of a variable index subset $\mathbb{M} \subseteq [D]$ (i.e., the $i$-th element of $g(\mathbb{M})$ is 1 if $i \in \mathbb{M}$, and 0 otherwise), and $/$ is the element-wise division. Each dimension of $\sum_{(\mathbb{M},\mathcal{D}) \in \mathbb{D}} \sum_{(\boldsymbol{x}^i, y^i) \in \mathcal{D}} y^i \cdot g(\mathbb{M})$ is the sum of query evaluations using each variable, and each dimension of $\sum_{(\mathbb{M},\mathcal{D}) \in \mathbb{D}} |\mathcal{D}| \cdot g(\mathbb{M})$ is the number of queries using each variable. Thus, the $i$-th element of variable score $\boldsymbol{s}$, representing the importance of the $i$-th variable $x_i$, is actually measured by the average goodness of all the sampled points that are generated by optimizing a subset of variables containing $x_i$. The variable score $\boldsymbol{s}$ will be used to define the value of each tree node of MCTS as well as for node expansion.

In MCTS-VS, the root of the tree represents all variables. A tree node $X$ represents a subset of variables, whose index set is denoted by $\mathbb{A}_X \subseteq [D]$, and it stores the value $v_X$ and the number $n_X$ of visits, which are used to calculate the value of UCB as in Eq. (1). The value $v_X$ is defined as the average score (i.e., importance) of the variables contained by $X$, which can be calculated by $\boldsymbol{s} \cdot g(\mathbb{A}_X)/|\mathbb{A}_X|$, where $g(\mathbb{A}_X)$ is the Boolean vector representation of $\mathbb{A}_X$ and $|\mathbb{A}_X|$ is the size of $\mathbb{A}_X$, i.e., the number of variables in node $X$.

At each iteration, MCTS-VS first recursively selects a node with larger UCB until a leaf node (denoted as $X$), which is regarded as containing important variables. Note that if we optimize the subset $\boldsymbol{x}_{\mathbb{A}_X}$ of variables represented by the leaf $X$ directly, the variables in $\boldsymbol{x}_{\mathbb{A}_X}$ will have the same score (because they are optimized together), and their relative importance cannot be further distinguished. Thus, MCTS-VS uniformly selects a variable index subset $\mathbb{M}$ from $\mathbb{A}_X$ at random, and employs BO to optimize $\boldsymbol{x}_{\mathbb{M}}$ as well as $\boldsymbol{x}_{\mathbb{A}_X \setminus \mathbb{M}}$; this process is repeated for several times. After that, the information set $\mathbb{D}$ will be augmented by the pairs of the selected variable index subset $\mathbb{M}$ (or $\mathbb{A}_X \setminus \mathbb{M}$) and the corresponding sampled points generated by BO. The variable score vector $\boldsymbol{s}$ will be updated using this new $\mathbb{D}$. Based on $\boldsymbol{s}$, the variable index set $\mathbb{A}_X$ represented by the leaf $X$ will be divided into two disjoint subsets, containing variables with larger and smaller scores (i.e., important and unimportant variables), respectively, and the leaf $X$ will be bifurcated into two child nodes accordingly. Finally, the $v$ values of these two children will be calculated using the variable score vector $\boldsymbol{s}$, and back-propagation will be performed to update the $v$ value and the number of visits of the nodes along the current path of the tree.

MCTS-VS can be equipped with any specific BO optimizer, resulting in the concrete algorithm MCTS-VS-BO, where BO is used to optimize the selected subsets of variables during the running of MCTS-VS. Compared with LA-MCTS [40], MCTS-VS applies MCTS to partition the variables instead of the search space, and thus can be more scalable. Compared with the previous variable selection method Dropout [21], MCTS-VS can select important variables automatically instead of randomly selecting a fixed number of variables in each iteration. Next we introduce it in detail.

### 3.1 Details of MCTS-VS

The procedure of MCTS-VS is described in Algorithm 1. In line 1, it first initializes the information set $\mathbb{D}$. In particular, a variable index subset $\mathbb{M}_i$ is randomly sampled from $[D]$, and the Latin hypercube sampling [24] is used to generate two sets (denoted as $\mathcal{D}_i$ and $\mathcal{D}_{\bar{i}}$) of $N_s$ points to form the two pairs of $(\mathbb{M}_i, \mathcal{D}_i)$ and $(\bar{\mathbb{M}}_i, \mathcal{D}_{\bar{i}})$, where $\bar{\mathbb{M}}_i = [D] \setminus \mathbb{M}_i$. This process will be repeated for $N_v$ times, resulting in the initial $\mathbb{D} = \{(\mathbb{M}_i, \mathcal{D}_i), (\bar{\mathbb{M}}_i, \mathcal{D}_{\bar{i}})\}_{i=1}^{N_v}$. The variable score vector $\boldsymbol{s}$ is calculated using this initial $\mathbb{D}$ in line 3, and the Monte Carlo tree is initialized in line 4 by adding only a root node, whose $v$ value is calculated according to $\boldsymbol{s}$ and number of visits is 0. MCTS-VS uses the variable $t$ to record the number of evaluations it has performed, and thus $t$ is set to $2 \times N_v \times N_s$ in line 5 as the initial $\mathbb{D}$ contains $2 \times N_v \times N_s$ sampled points in total.

In each iteration (i.e., lines 7–28) of MCTS-VS, it selects a leaf node $X$ by UCB in line 10, and optimizes the variables (i.e., $\boldsymbol{x}_{\mathbb{A}_X}$) represented by $X$ in lines 13–23. Note that to measure the relative importance of variables in $\boldsymbol{x}_{\mathbb{A}_X}$, MCTS-VS optimizes different subsets of variables of $\boldsymbol{x}_{\mathbb{A}_X}$

**Algorithm 1** MCTS-VS

**Parameters**: batch size $N_v$ of variable index subset, sample batch size $N_s$, total number $N_e$ of evaluations, threshold $N_{bad}$ for re-initializing a tree and $N_{split}$ for splitting a node, hyper-parameter $k$ for the best-$k$ strategy

**Process**:

1: Initialize the information set $\mathbb{D} = \{(\mathbb{M}_i, \mathcal{D}_i), (\bar{\mathbb{M}}_i, \mathcal{D}_{\bar{i}})\}_{i=1}^{N_v}$;
2: Store the best $k$ sampled points in $\mathbb{D}$;
3: Calculate the variable score $\boldsymbol{s}$ using $\mathbb{D}$ as in Eq. (2);
4: Initialize the Monte Carlo tree;
5: Set $t = 2 \times N_v \times N_s$ and $n_{bad} = 0$;
6: **while** $t < N_e$ **do**
7:    **if** $n_{bad} > N_{bad}$ **then**
8:       Initialize the Monte Carlo tree and set $n_{bad} = 0$
9:    **end if**
10:    $X \leftarrow$ the leaf node selected by UCB;
11:    Let $\mathbb{A}_X$ denote the indexes of the subset of variables represented by $X$;
12:    Increase $n_{bad}$ by 1 once visiting a right child node on the path from the root node to $X$;
13:    **for** $j = 1 : N_v$ **do**
14:       Sample a variable index subset $\mathbb{M}$ from $\mathbb{A}_X$ uniformly at random;
15:       Fit a GP model using the points $\{(\boldsymbol{x}_{\mathbb{M}}^i, y^i)\}_{i=1}^t$ sampled-so-far, where only the variables indexed by $\mathbb{M}$ are used;
16:       Generate $\{\boldsymbol{x}_{\mathbb{M}}^{t+i}\}_{i=1}^{N_s}$ by maximizing an acquisition function;
17:       Determine $\{\boldsymbol{x}_{[D]\backslash\mathbb{M}}^{t+i}\}_{i=1}^{N_s}$ by the "fill-in" strategy;
18:       Evaluate $\boldsymbol{x}^{t+i} = [\boldsymbol{x}_{\mathbb{M}}^{t+i}, \boldsymbol{x}_{[D]\backslash\mathbb{M}}^{t+i}]$ to obtain $y^{t+i}$ for $i = 1, 2, \ldots, N_s$;
19:       $\mathbb{D} = \mathbb{D} \cup \{(\mathbb{M}, \{(\boldsymbol{x}^{t+i}, y^{t+i})\}_{i=1}^{N_s})\}$;
20:       Store the best $k$ points sampled-so-far;
21:       $t = t + N_s$;
22:       Repeat lines 15–21 for $\bar{\mathbb{M}} = \mathbb{A}_X \backslash \mathbb{M}$
23:    **end for**
24:    Calculate the variable score $\boldsymbol{s}$ using $\mathbb{D}$ as in Eq. (2);
25:    **if** $|\mathbb{A}_X| > N_{split}$ **then**
26:       Bifurcate the leaf node $X$ into two child nodes, whose $v$ value and number of visits are calculated by $\boldsymbol{s}$ and set to 0, respectively
27:    **end if**
28:    Back-propagate to update the $v$ value and number of visits of the nodes on the path from the root to $X$
29: **end while**

---

instead of $\boldsymbol{x}_{\mathbb{A}_X}$ directly. That is, a variable index subset $\mathbb{M}$ is randomly sampled from $\mathbb{A}_X$ in line 14, and the corresponding subset $\boldsymbol{x}_{\mathbb{M}}$ of variables is optimized by BO in lines 15–16. The data points $\{(\boldsymbol{x}_{\mathbb{M}}^i, y^i)\}_{i=1}^t$ sampled-so-far is used to fit a GP model, and $N_s$ (called *sample batch size*) new points $\{\boldsymbol{x}_{\mathbb{M}}^{t+i}\}_{i=1}^{N_s}$ are generated by maximizing an acquisition function. Note that this is a standard BO procedure, which can be replaced by any other variant. To evaluate $\boldsymbol{x}_{\mathbb{M}}^{t+i}$, we need to fill in the values of the other variables $\boldsymbol{x}_{[D]\backslash\mathbb{M}}^{t+i}$, which will be explained later. After evaluating $\boldsymbol{x}^{t+i} = [\boldsymbol{x}_{\mathbb{M}}^{t+i}, \boldsymbol{x}_{[D]\backslash\mathbb{M}}^{t+i}]$ in line 18, the information set $\mathbb{D}$ is augmented with the new pair of $(\mathbb{M}, \{(\boldsymbol{x}^{t+i}, y^{t+i})\}_{i=1}^{N_s})$ in line 19, and $t$ is increased by $N_s$ accordingly in line 21. For fairness, the complement subset $\boldsymbol{x}_{\bar{\mathbb{M}}}$ of variables, where $\bar{\mathbb{M}} = \mathbb{A}_X \backslash \mathbb{M}$, is also optimized by the same way, i.e., lines 15–21 of Algorithm 1 is repeated for $\bar{\mathbb{M}}$. The whole process of optimizing $\boldsymbol{x}_{\mathbb{M}}$ and $\boldsymbol{x}_{\bar{\mathbb{M}}}$ in lines 14–22 will be repeated for $N_v$ times, which is called *batch size of variable index subset*.

To fill in the values of the un-optimized variables in line 17, we employ the *best-$k$* strategy, which utilizes the best $k$ data points sampled-so-far, denoted as $\{(\boldsymbol{x}^{*j}, y^{*j})\}_{j=1}^k$. That is, $\{y^{*j}\}_{j=1}^k$ are the $k$ largest objective values observed-so-far. If the $i$-th variable is un-optimized, its value will be uniformly selected from $\{x_i^{*j}\}_{j=1}^k$ at random. Thus, MCTS-VS needs to store the best $k$ data points in line 2 after initializing the information set $\mathbb{D}$, and update them in line 20 after augmenting $\mathbb{D}$. Other direct "fill-in" strategies include sampling the value randomly, or using the average variable

value of the best $k$ data points. The superiority of the employed best-$k$ strategy will be shown in the experiments in Appendix D.

After optimizing the variables $\boldsymbol{x}_{\mathbb{A}_X}$ represented by the selected leaf $X$, the variable score vector $\boldsymbol{s}$ measuring the importance of each variable will be updated using the augmented $\mathbb{D}$ in line 24. If the number $|\mathbb{A}_X|$ of variables in the leaf $X$ is larger than a threshold $N_{split}$ (i.e., line 25), $\mathbb{A}_X$ will be divided into two subsets. One contains those "important" variable indexes with score larger than the average score of $\boldsymbol{x}_{\mathbb{A}_X}$, and the other contains the remaining "unimportant" ones. The leaf $X$ will be bifurcated into a left child $Y$ and a right child $Z$ in line 26, containing those important and unimportant variables, respectively. Meanwhile, $v_Y$ and $v_Z$ will be calculated according to $\boldsymbol{s}$, and the number of visits is 0, i.e., $n_Y = n_Z = 0$. Finally, MCTS-VS performs back-propagation in line 28 to re-calculate the $v$ value and increase the number of visits by 1 for each ancestor of $Y$ and $Z$.

MCTS-VS will run until the number $t$ of performed evaluations reaches the budget $N_e$. Note that as the Monte Carlo tree may be built improperly, we use a variable $n_{bad}$ to record the number of visiting a right child node (regarded as containing unimportant variables), measuring the goodness of the tree. In line 5 of Algorithm 1, $n_{bad}$ is initialized as 0. During the procedure of selecting a leaf node by UCB in line 10, $n_{bad}$ will be increased by 1 once visiting a right child node, which is updated in line 12. If $n_{bad}$ is larger than a threshold $N_{bad}$ (i.e., line 7), the current tree is regarded as bad, and will be re-initialized in line 8. Furthermore, the frequency of re-initialization can be used to indicate whether MCTS-VS can do a good variable selection for the current problem. For ease of understanding, we also provide an example illustration of MCTS-VS in Appendix A.

## 4 Theoretical Analysis

Although it is difficult to analyze the regret of MCTS-VS directly, we can theoretically analyze the influence of general variable selection by adopting the acquisition function GP-UCB. The considered general variable selection framework is as follows: after selecting a subset of variables at each iteration, the corresponding observation data (i.e., the data points sampled-so-far where only the selected variables are used) is used to build a GP model, and the next data point is sampled by maximizing GP-UCB. We use $\mathbb{M}_t$ to denote the sampled variable index subset at iteration $t$, and let $|\mathbb{M}_t| = d_t$.

**Regret Analysis.** Let $\boldsymbol{x}^*$ denote an optimal solution. We analyze the cumulative regret $R_T = \sum_{t=1}^{T}(f(\boldsymbol{x}^*) - f(\boldsymbol{x}^t))$, i.e., the sum of the gap between the optimum and the function values of the selected points by iteration $T$. To derive an upper bound on $R_T$, we pessimistically assume that the worst function value, i.e., $\min_{\boldsymbol{x}_{[D]\setminus\mathbb{M}_t}} f([\boldsymbol{x}_{\mathbb{M}_t}, \boldsymbol{x}_{[D]\setminus\mathbb{M}_t}])$, given $\boldsymbol{x}_{\mathbb{M}_t}$ is returned in evaluation. As in [21, 38], we assume that $\mathcal{X} \subset [0, r]^D$ is convex and compact, and $f$ satisfies the following Lipschitz assumption.

**Assumption 4.1.** The function $f$ is a GP sample path. For some $a, b > 0$, given $L > 0$, the partial derivatives of $f$ satisfy that $\forall i \in [D], \exists \alpha_i \geq 0$,
$$P\left(\sup_{\boldsymbol{x}\in\mathcal{X}} |\partial f/\partial x_i| < \alpha_i L\right) \geq 1 - ae^{-(L/b)^2}. \tag{3}$$
Based on Assumption 4.1, we define $\alpha_i^*$ to be the minimum value of $\alpha_i$ such that Eq. (3) holds, which characterizes the importance of the $i$-th variable $x_i$. The larger $\alpha_i^*$, the greater influence of $x_i$ on the function $f$. Let $\alpha_{\max} = \max_{i\in[D]} \alpha_i^*$.

Theorem 4.2 gives an upper bound on the cumulative regret $R_T$ with high probability for general variable selection methods. The proof is inspired by that of GP-UCB without variable selection [38] and provided in Appendix B.1. If we select all variables each time (i.e., $\forall t : \mathbb{M}_t = [D]$) and assume $\forall i : \alpha_i^* \leq 1$, the regret bound Eq. (4) becomes $R_T \leq \sqrt{C_1 T \beta_T^* \gamma_T} + 2$, which is consistent with [38]. Note that $\forall t : |\mathbb{M}_t| = d_t = D$ in this case, which implies that $\beta_t$ increases with $t$, leading to $\beta_T^* = \beta_T$. We can see that using variable selection will increase $R_T$ by $2\sum_{t=1}^{T}\sum_{i\in[D]\setminus\mathbb{M}_t} \alpha_i^* L r$, related to the importance (i.e., $\alpha_i^*$) of unselected variables at each iteration. The more important variables unselected, the larger $R_T$. Meanwhile, the term $\sqrt{C_1 T \beta_T^* \gamma_T}$ will decrease as $\beta_T^*$ relies on the number $d_t$ of selected variables positively. Ideally, if the unselected variables at each iteration are always unrelated (i.e., $\alpha_i^* = 0$), the regret bound will be better than that of using all variables [38].

**Theorem 4.2.** $\forall \delta \in (0, 1)$, *let* $\beta_t = 2\log(4\pi_t/\delta) + 2d_t \log(d_t t^2 br\sqrt{\log(4Da/\delta)})$ *and* $L = b\sqrt{\log(4Da/\delta)}$, *where* $r$ *is the upper bound on each variable, and* $\{\pi_t\}_{t\geq 1}$ *satisfies* $\sum_{t\geq 1} \pi_t^{-1} = 1$

*and $\pi_t > 0$. Let $\beta_T^* = \max_{1 \le i \le T} \beta_t$. At iteration $T$, the cumulative regret*

$$R_T \le \sqrt{C_1 T \beta_T^* \gamma_T} + 2\alpha_{\max} + 2\sum_{t=1}^{T}\sum_{i \in [D] \backslash \mathbb{M}_t} \alpha_i^* L r \tag{4}$$

*holds with probability at least $1-\delta$, where $C_1$ is a constant, $\gamma_T = \max_{|\mathcal{D}|=T} I(\boldsymbol{y}_{\mathcal{D}}, \boldsymbol{f}_{\mathcal{D}})$, $I(\cdot, \cdot)$ is the information gain, and $\boldsymbol{y}_{\mathcal{D}}$ and $\boldsymbol{f}_{\mathcal{D}}$ are the noisy and true observations of a set $\mathcal{D}$ of points, respectively.*

By selecting $d$ variables randomly at each iteration and assuming that $r = 1$ and $\forall i : \alpha_i^* \le 1$, it has been proved [21] that the cumulative regret of Dropout satisfies

$$R_T \le \sqrt{C_1 T \beta_T \gamma_T} + 2 + 2TL(D - d). \tag{5}$$

In this case, we have $d_t = |\mathbb{M}_t| = d$, $r = 1$ and $\forall i : \alpha_i^* \le 1$. Thus, Eq. (4) becomes

$$R_T \le \sqrt{C_1 T \beta_T^* \gamma_T} + 2 + 2TL(D - d). \tag{6}$$

Note that $\beta_T^* = \beta_T$ here, as $\beta_t$ increases with $t$ given $d_t = d$. This implies that our bound Eq. (4) for general variable selection is a generalization of Eq. (5) for Dropout [21]. In [33], a regret bound analysis has also been performed for variable selection, by optimizing over $d$ fixed important variables and using a common parameter $\alpha$ to characterize the importance of all the other $D - d$ variables.

**Computational Complexity Analysis.** The computational complexity of one iteration of BO depends on three critical components: fitting a GP surrogate model, maximizing an acquisition function and evaluating a sampled point. If using the squared exponential kernel, the computational complexity of fitting a GP model at iteration $t$ is $\mathcal{O}(t^3 + t^2 d_t)$. Maximizing an acquisition function is related to the optimization algorithm. If we use the Quasi-Newton method to optimize GP-UCB, the computational complexity is $\mathcal{O}(m(t^2 + td_t + d_t^2))$ [28], where $m$ denotes the Quasi-Newton's running rounds. The cost of evaluating a sampled point is fixed. Thus, by selecting only a subset of variables, instead of all variables, to optimize, the computational complexity can be decreased significantly. The detailed analysis is provided in Appendix B.2.

**Insight.** The above regret and computational complexity analyses have shown that variable selection can reduce the computational complexity while increasing the regret. Given the number $d_t$ of variables to be selected, a good variable selection method should select as important variables as possible, i.e., variables with as large $\alpha_i^*$ as possible, which may help to design and evaluate variable selection methods. The experiments in Section 5.1 will show that MCTS-VS can select a good subset of variables while maintaining a small computational complexity.

## 5   Experiment

To examine the performance of MCTS-VS, we conduct experiments on different tasks, including synthetic functions, NAS-bench problems and MuJoCo locomotion tasks, to compare MCTS-VS with other black-box optimization methods. For MCTS-VS, we use the same hyper-parameters except $C_p$, which is used for calculating UCB in Eq. (1). For Dropout and embedding-based methods, we set the parameter $d$ to the number of valid dimensions for synthetic functions, and a reasonable value for real-world problems. The hyper-parameters of the same components of different methods are set to the same. We use five identical random seeds (2021–2025) for all problems and methods. More details about the settings can be found in Appendix C. Our code is available at `https://github.com/lamda-bbo/MCTS-VS`.

### 5.1   Synthetic Functions

We use Hartmann ($d = 6$) and Levy ($d = 10$) as the synthetic benchmark functions, and extend them to high dimensions by adding unrelated variables as [20, 27, 42]. For example, Hartmann6_300 has the dimension $D = 300$, and is generated by appending $294$ unrelated dimensions to Hartmann. The variables affecting the value of $f$ are called *valid variables*.

**Effectiveness of Variable Selection.** Dropout [21] is the previous variable selection method which randomly selects $d$ variables in each iteration, while our proposed MCTS-VS applies MCTS to automatically select important variables. We compare them against vanilla BO [32] without variable selection. The first two subfigures in Figure 1 show that Dropout-BO and MCTS-VS-BO are better than vanilla BO, implying the effectiveness of variable selection. We can also see that MCTS-VS-BO performs the best, implying the superiority of MCTS-based variable selection over random selection.

We also equip MCTS-VS and Dropout with the advanced BO algorithm TuRBO [10], resulting in MCTS-VS-TuRBO and Dropout-TuRBO. The last two subfigures in Figure 1 show the similar results except that MCTS-VS-TuRBO needs more evaluations to be better than Dropout-TuRBO. This is because TuRBO costs more evaluations than BO on the same selected variables, and thus needs more evaluations to generate sufficient samples for an accurate estimation of the variable score in Eq. (2).

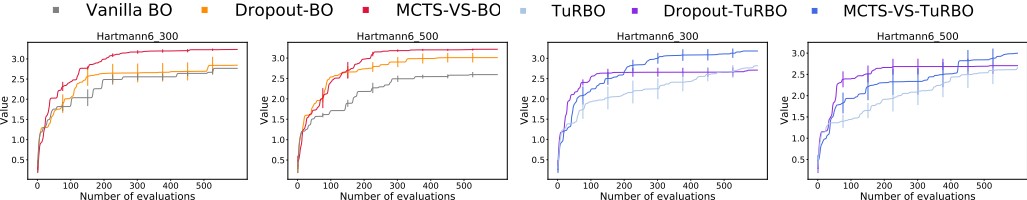

Figure 1: Performance comparison among the two variable selection methods (i.e., MCTS-VS and Dropout) and the BO methods (i.e., Vanilla BO and TuRBO) on two synthetic functions.

**Comparison with State-of-The-Art Methods.** We compare MCTS-VS with the state-of-the-art methods, including TuRBO [10], LA-MCTS-TuRBO [40], SAASBO [9], HeSBO [27], ALEBO [20] and CMA-ES [14]. TuRBO fits a collection of local models to optimize in the trust regions for overcoming the homogeneity of the global model and over-exploration. LA-MCTS-TuRBO applies MCTS to partition the search space and uses TuRBO to optimize in a small sub-region. SAASBO uses sparsity-inducing prior to select variables implicitly. HeSBO and ALEBO are state-of-the-art embedding methods. CMA-ES is a popular evolutionary algorithm. We also implement VAE-BO by combining VAE [18] with vanilla BO directly, as a baseline of learning-based embedding. For MCTS-VS, we implement the two versions of MCTS-VS-BO and MCTS-VS-TuRBO, i.e., MCTS-VS equipped with vanilla BO and TuRBO.

As shown in Figure 2, MCTS-VS can achieve the best performance except on Levy10_100, where it is a little worse than TuRBO. For low-dimensional functions (e.g., $D = 100$ for Levy10_100), TuRBO can adjust the trust region quickly while MCTS-VS needs samples to estimate the variable score. But as the dimension increases, the search space increases exponentially and it becomes difficult for TuRBO to adjust the trust region; while the number of variables only increases linearly, making MCTS-VS more scalable. SAASBO has similar performance to MCTS-VS due to the advantage of sparsity-inducing prior. HeSBO is not stable, which has a moderate performance on Hartmann but a relatively good performance on Levy. Note that we only run SAASBO and ALEBO for 200 evaluations on Hartmann functions because it has already taken more than hours to finish one iteration when the number of samples is large. More details about runtime are shown in Table 1. VAE-BO has the worst performance, suggesting that the learning algorithm in high-dimensional BO needs to be designed carefully. We also conduct experiments on extremely low and high dimensional variants of Hartmann (i.e., Hartmann6_100 and Hartmann6_1000), showing that MCTS-VS still performs well, and perform the significance test by running each method more times. Please see Appendix E.

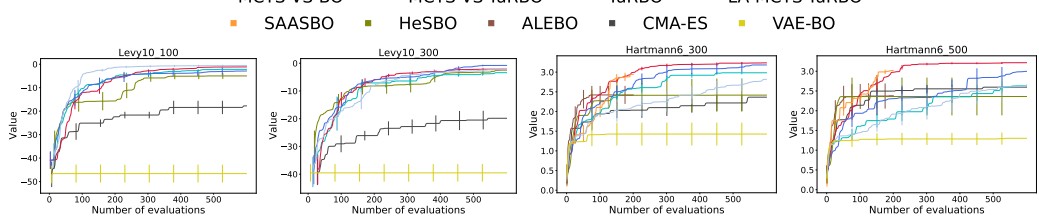

Figure 2: Comparison among MCTS-VS and state-of-the-art methods on synthetic functions.

Next, we compare the practical running overheads of these methods. We run each method for 100 evaluations independently using 30 different random seeds, and calculate the average wall clock time. The results are shown in Table 1. As expected, when using variable selection (i.e., Dropout and MCTS-VS), the time is less than that of Vanilla BO or TuRBO, because we only optimize a subset of variables. MCTS-VS is a little slower than Dropout, which is because MCTS-VS needs to build the search tree and calculate the variable score, while Dropout only randomly selects variables. MCTS-VS is much faster than LA-MCTS-TuRBO, showing the advantage of partitioning the variables to partitioning the search space. SAASBO optimizes all variables instead of only a subset of variables and uses No-U-Turn sampler (NUTS) to inference, consuming $\times 500 - \times 1000$ time. HeSBO and

Table 1: Wall clock time (in seconds) comparison among different methods.

| METHOD | LEVY10_100 | LEVY10_300 | HARTMANN6_300 | HARTMANN6_500 |
|---|---|---|---|---|
| VANILLA BO | 3.190 | 4.140 | 4.844 | 5.540 |
| DROPOUT-BO | 2.707 | 3.225 | 3.237 | 3.685 |
| MCTS-VS-BO | 2.683 | 3.753 | 3.711 | 4.590 |
| TURBO | 8.621 | 9.206 | 9.201 | 9.754 |
| LA-MCTS-TURBO | 14.431 | 22.165 | 25.853 | 34.381 |
| MCTS-VS-TURBO | 4.912 | 5.616 | 5.613 | 5.893 |
| SAASBO | / | / | 2185.678 | 4163.121 |
| HESBO | 220.459 | 185.092 | 51.678 | 55.699 |
| ALEBO | / | / | 470.714 | 512.641 |
| CMA-ES | 0.030 | 0.043 | 0.043 | 0.045 |

Table 2: Recall comparison between MCTS-VS and Dropout.

| METHOD | LEVY10_100 | LEVY10_300 | HARTMANN6_300 | HARTMANN6_500 |
|---|---|---|---|---|
| DROPOUT | 0.100 | 0.030 | 0.020 | 0.012 |
| MCTS-VS | 0.429 | 0.433 | 0.352 | 0.350 |

ALEBO consume $\times 10 - \times 500$ time compared with the variable selection methods. CMA-ES is very fast because it does not need to fit a GP model or optimize an acquisition function. The reasons for the small running overhead of MCTS-VS can be summarized as follows: 1) it only optimizes a selected subset of variables; 2) the depth of the search tree is shallow, i.e., $O(\log D)$ in expectation and less than $D$ in the worse case; 3) the variable score vector in Eq. (2) is easy to calculate for bifurcating a tree node.

**Why MCTS-VS Can Perform Well.** The theoretical results have suggested that a good variable selection method should select as important variables as possible. Thus, we compare the quality of the variables selected by MCTS-VS and Dropout (i.e., random selection), measured by the recall $d_t^*/d$, where $d$ is the number of valid variables, and $d_t^*$ is the number of valid variables selected at iteration $t$. Dropout randomly selects $d$ variables at each iteration, and thus, the recall is $d/D$ in expectation. For MCTS-VS, we run MCTS-VS-BO for $600$ evaluations on five different random seeds, and calculate the average recall. As shown in Table 2, the average recall of MCTS-VS is much larger than that of Dropout, implying that MCTS-VS can select better variables than random selection, and thus achieve a good performance as shown before. Meanwhile, the recall between $0.35$ and $0.433$ of MCTS-VS also implies that the variable selection method could be further improved.

## 5.2 Real-World Problems

We further compare MCTS-VS with the baselines on real-world problems, including NAS-Bench-101 [44], NAS-Bench-201 [7], Hopper and Walker2d. NAS-Bench problems are popular benchmarks in high-dimensional BO. Hopper and Walker2d are robot locomotion tasks in MuJoCo [39], which is a popular black-box optimization benchmark and much more difficult than NAS-Bench problems. The experimental results on more real-world problems can refer to Appendix E.

**NAS-Bench Problems.** NAS-Bench-101 is a tabular data set that maps convolutional neural network architectures to their trained and evaluated performance on CIFAR-10, and we create a constrained problem with $D = 36$ in the same way as [20]. NAS-Bench-201 is an extension to NAS-Bench-101, leading to a problem with $D = 30$ but without constraints. Figure 3 shows the results with the wall clock time as the $x$-axis, where the gray dashed line denotes the optimum. The results using the number of evaluations as the $x$-axis are provided in Appendix E, showing that the performance of BO-style methods is similar, as already observed in [20]. This may be because there are many structures whose objective values are close to the optimum. But when considering the actual runtime, MCTS-VS-BO is still clearly better as shown in Figure 3, due to the advantage of variable selection. We also provide results on more NAS-Bench problems, including NAS-Bench-1Shot1 [45], TransNAS-Bench-101 [8] and NAS-Bench-ASR [25] in Appendix E.

**MuJoCo Locomotion Tasks.** Next we turn to the more difficult MuJoCo tasks in RL. The goal is to find the parameters of a linear policy maximizing the accumulative reward. Different from

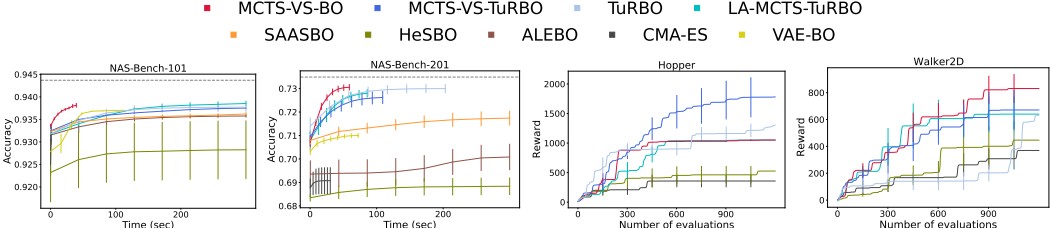

Figure 3: Comparison on NAS-Bench.  Figure 4: Comparison on MuJoCo.

previous problems, the objective $f$ (i.e., the accumulative reward) is highly stochastic here, making it difficult to solve. We use the mean of three independent evaluations to estimate $f$, and limit the evaluation budget to 1200 due to expensive evaluation. Note that we do not run SAASBO, ALEBO, and VAE-BO because SAASBO and ALEBO are extremely time-consuming, and VAE-BO behaves badly in previous experiments. The results are shown in Figure 4. TuRBO behaves well on Hopper with a low dimension $D = 33$, and MCTS-VS-TuRBO, combining the advantage of variable selection and TuRBO, achieves better performance, outperforming all the other baselines. On Walker2d with a high dimension $D = 102$, MCTS-VS-BO performs the best, because of the good scalability. Most methods have large variance due to the randomness of $f$. For HeSBO, we have little knowledge about the parameter $d$, and use 10 and 20 for Hopper and Walker2d, respectively. Its performance may be improved by choosing a better $d$, which, however, requires running the experiment many times, and is time-consuming. Note that on the two MuJoCo tasks, Hopper and Walker2d, each variable is valid. The good performance of MCTS-VS may be because optimizing only a subset of variables is sufficient for achieving the goal and MCTS-VS can select them. For example, the Walker2D robot consists of four main body parts: a torso, two thighs, two legs and two feet, where adjacent ones are connected by two hinges. The goal is to move forward by optimizing the hinges, each of which is valid. But even locking the hinges between legs and feet, the robot can still move forward by optimizing the other hinges. This is similar to that when the ankles are fixed, a person can still walk.

**Further Studies.** We further perform sensitivity analysis about the hyper-parameters of MCTS-VS, including the employed optimizer, "fill-in" strategy, $C_p$ for calculating UCB in Eq. (1), number $2 \times N_v \times N_s$ of sampled data in each iteration, threshold $N_{bad}$ for re-initializing a tree and $N_{split}$ for splitting a tree node. Please see Appendix D. We conduct additional experiments in Appendix E, including experiments on synthetic functions depending on a subset of variables to various extent and with increasing ratio of valid variables, examination of combining MCTS-VS with SAASBO (which can be viewed as a hierarchical variable selection method), and comparison with other variable selection methods (e.g., LASSO).

# 6    Conclusion

In this paper, we propose the MCTS-VS method for variable selection in high-dimensional BO, which uses MCTS to recursively partition the variables into important and unimportant ones, and only optimizes those important variables. Theoretical results suggest selecting as important variables as possible, which may be of independent interest for variable selection. Comprehensive experiments on synthetic, NAS-bench and MuJoCo problems demonstrate the effectiveness of MCTS-VS.

However, MCTS-VS relies on the assumption of low effective dimensionality, and might not work well if the percentage of valid variables is high. The amount of hyper-parameters might be another limitation, though our sensitivity analysis has shown that the performance of MCTS-VS is not sensitive to most hyper-parameters. The current theoretical analysis is for general variable selection, while it will be very interesting to perform specific theoretical analysis for MCTS-VS.

### Acknowledgement

The authors would like to thank reviewers for their helpful comments and suggestions. This work was supported by the NSFC (62022039, 62276124) and the Fundamental Research Funds for the Central Universities (0221-14380014).

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
