# OpenReview forum: "Monte Carlo Tree Search based Variable Selection for High Dimensional Bayesian Optimization"
_NeurIPS.cc/2022/Conference — NeurIPS 2022 Accept_

### Official Review · Reviewer_do8w · 2022-07-06

**Rating:** 8
**Confidence:** 4
**Soundness:** 4 excellent
**Presentation:** 4 excellent
**Contribution:** 3 good

**Summary:**

This paper studies the high-dimensional black-box optimization problem and proposes a general variable selection method based on Monte Carlo Tree Search (MCTS) for Bayesian optimization. The authors performed comprehensive experiments to show the effectiveness of the proposed method, as well as theoretical analyses to explain why their method can be better.

**Questions:**

1. For the unselected variables, you randomly sampled their values from the best $k$ generated data points. The experiments have shown that this strategy is better than selecting the values completely at random. But I also would like to see how about using the average of the best $k$ data points.

2. In figure 14, why is HeSBO not compared?

**Limitations:**

Yes.

**Strengths And Weaknesses:**

As far as I know, Dropout [Li et al., 2017] is the only variable selection method for high-dimensional BO. This paper proposes another novel one using MCTS.

The main idea is to apply MCTS to select a subset of important variables for optimization in each iteration. A key component of their method is the introduced concept of the variable score, which measures the importance of each variable and is used for node selection and expansion in MCTS. They estimate the score of one variable using the objective values of the data points sampled by optimizing any subset of variables containing it. The idea is natural and reasonable and is expected to be superior to Dropout (random selection), which has been validated by the authors’ experiments.

The most related method I know is LA-MCTS [Wang et al., 2020], which uses MCTS to divide the range of variables (not to decrease the dimension), and optimizes in a sub-region of the search space. The proposed method so-called MCTS-VS in this paper uses MCTS to select a subset of variables, having better scalability to high-dimensional problems. The authors have made a good explanation of their difference.

The authors did comprehensive experiments. They tested the algorithms on common synthetic functions (their various high-dimensional variants), and various real-world problems, including five NAS-Bench problems (three provided in the appendix) and two MuJoCo tasks in reinforcement learning. The results show that MCTS-VS is superior to Dropout for variable selection, and achieves better performance than SOTA BO methods. They also analyzed the sensitivity of various parameters, giving some guidance on using MCTS-VS in practice. The experiments are convincing to me.

The authors also analyzed the regret bound of variable selection methods. The proofs are non-trivial, but technically not very novel, which is mainly based on [Srinivas et al. 2012]. The derived regret bound is a generalization of GP-UCB [Srinivas et al. 2012] and Dropout [Li et al., 2017]. The theoretical part is not very impressive, but OK, giving some insight into how variable selection methods work. Based on the theoretical findings, the authors have empirically explained why MCTS-CS can be better, by showing that MCTS-CS can select much more significant variables than random selection (Dropout) in the experiments. I suggest the authors to consider specific regret analysis about MCTS-VS, which though seems to be very challenging.

I have read the paper carefully, including the appendix. I did not find any mistakes in their proofs. The authors have described the experiments in sufficient detail, which I believe are reproducible. They also provided the codes in the supplementary material.

The paper is very well written and organized. The method is presented very clearly and is easy to follow. I appreciate that the authors provided an example illustrated in the appendix.

Overall, I think this is a very good work, studying the important high-dimensional black-box optimization problem, proposing a novel variable selection method for BO, performing theoretical analyses (not very impressive, but OK), and showing the effectiveness of the proposed method by very comprehensive experiments. I believe this work would be a good resource to the high-dimensional BO community, which may motivate more variable selection works.

Some minor comments:
-L56, you should give the full name when first mentioning the abbreviation PI, EI and UCB.

-caption of figure 1, “four synthetic functions” -> “two synthetic functions”

-It needs to explain “recall comparison” more clearly in the caption of Table 2, though I see the explanation in the paper.

-L563 and 580, “a confidence bound of” -> “a confidence bound on”

-In the equations between L651 and 652, you should clearly indicate that $|\cdot|$ denotes the determinant of a matrix, and $tr(\cdot)$ denotes the trace of a matrix.

-L738, the right brackets is missing.

-On top of figure 9, “C_p” (e.g., in Levy10_100_C_p) is redundant.

-L790, “the synthetic functions” -> “a synthetic function”

-L796, “tes ts” -> “tests”

---

> ### Author Response · Authors · 2022-08-02
> **Response to Reviewer do8w**
>
> Thank you for your positive review. We are very grateful to you for carefully reviewing our paper and providing constructive comments and suggestions. Below please find our response.
>
> Q1: "For the unselected variables, you randomly sampled their values from the best $k$ generated data points. ... But I also would like to see how about using the average of the best $k$ data points."
>
> R1: Thanks to your suggestion, we have added the comparison with the strategy of using the average of the best $k$ data points. Please see Figure 8(a) in the Appendix D of the revised version. The results show that the performance of the average strategy is between the best-$k$ and random strategies. Thus, the best-$k$ strategy is still recommended in practice.
>
> Q2: "In figure 14, why is HeSBO not compared?"
>
> R2: Thank you for pointing out this issue. We are very sorry that we missed HeSBO in Figure 14. We have now revised to add it. The results show that HeSBO has similar performance to LA-MCTS-TuRBO and is worse than MCTS-VS and TuRBO.
>
> Suggestion: "I suggest the authors to consider specific regret analysis about MCTS-VS, which though seems to be very challenging."
>
> R: Thank you for your suggestion. We fully agree that it is very interesting to perform specific regret analysis about MCTS-VS, which is also suggested by Reviewers EcyS and SbWH. One possible way is to analyze the bound on the number of unselected important variables using the UCB-based node selection of MCTS, and then combine it with the general bound in the paper. To be honest, we tried to perform this specific analysis, but can not finish it currently. In the last paragraph of Conclusion, we have revised to clearly indicate this limitation and leave it as our future work.
>
> For the minor comments, we have revised all of them in the new version. Please also note that to shorten the space, we have deleted Table 2, and the corresponding experimental results are presented in the last paragraph of Section 5.1. Thank you very much.

---

> > ### Comment · Reviewer_do8w · 2022-08-07
> > **Response to Authors**
> >
> > I have read all the reviews and the responses. I have also checked the revisions highlighted in the paper. Thank you very much for addressing the questions so detailed. I am satisfied with the revision, and I believe this work will be a good resource for high-dimensional BO.

---

### Official Review · Reviewer_SbWH · 2022-07-10

**Rating:** 6
**Confidence:** 3
**Soundness:** 2 fair
**Presentation:** 3 good
**Contribution:** 3 good

**Summary:**

This paper proposes a high-dimensional Bayesian optimization adopting Monte Carlo Tree Search as a variable selector. The competence of the proposed details of MCTS is demonstrated by consistent performance in a range of experiments.

**Questions:**

* The effectiveness of MCTS design is well-demonstrated in the experiment. However, some design choices may raise questions to some readers. The critical and novel component of MCTS is 'the variable score' in eq.(2). As explained, it is the sum of query evaluations using given variables. And I can imagine some scenarios where the proposed variable score may not behave as expected.
  * When the objective is positive and the task is maximization, in one extreme, if one variable appears in many subsets then it will have a high score not because it contributes to the optimization but because it just appears more often.
  * Another case is when the objective is negative and the task is maximization, even when a variable contributes to the optimization, by the design philosophy of the variable score, it should be evaluated more often, but due to its negative value, more addition will decrease its variable score.

> In these cases, how does the variable score behave? Or the objectives are assumed to be positive? Please correct me if there is a misconception.

* It seems that $r$ in Theorem 4.2 appearing in $\beta_t$ is another hyperparameter and any comment on this is missing. Can any comment be added?

**Ethics Review Area:**

["I don’t know"]

**Limitations:**

* Even though the provided regret analysis is, in a sense, novel, which allows us to get a sense of the effect of variable selection methods, this bound is quite loose. By using $\min_{i \in [D]} \alpha_i^*$, the right hand side in eq.(4) is lower bounded with a formula with a term linear w.r.t $T$. Therefore, this bound does not tell us the behavior of interest (sublinear). Following the argument in the last paragraph of Section 4, with this bound, what we can say at most is that a better variable selection method reduces the cumulative regret but it is guaranteed only up to having linear regret. Even though a good variable selection method seems helpful, the theoretical argument with this linear in $T$ bound is a bit insufficient. However, this may be a good intermediate step to the bound which tells us the effect of variable selection methods in making the cumulative regret sublinear. It would be better if this point can be addressed.

* The method relies on the assumption of 'low effective dimensionality. The strength of the method is to learn that low dimensions (subset of variables). The benchmarks are such function of low effective dimensionality and NAS-bench is used to show the computational efficiency of MCTS-VS without losing optimization performance. I admit that the MoJuCo experiments support that MCTS-VS may perform well for the function without low effective dimensionality. Still, it would be interesting to see the performance of MCTS-VS on some problems which are known not to have low effective dimensionality. A discussion on this can be addressed in a revised version.


**Strengths And Weaknesses:**

Strengths
* MCTS is effectively designed and used for the variable selection for high dimensional BO.
* The proposed algorithm exhibits consistent performance on various problems, demonstrating the effectiveness of the detailed design of MCTS.
* The experiments are conducted from various angles, and many ablation studies seem to be performed adequately.
* The algorithm is clearly explained.

Weaknesses
* The rationale behind some design choices is a bit unclear (detailed below)
* The provided theoretical bound only guarantees that a better variable selection method reduces the cumulative regret but leaves it as a linear regret (detailed below)

---

> ### Author Response · Authors · 2022-08-02
> **Response to Reviewer SbWH (1/2)**
>
> Thank you for your positive review. We are very grateful to you for carefully reviewing our paper and providing constructive comments and suggestions. Below please find our response.
>
> Q1: "The critical and novel component of MCTS is 'the variable score' in eq.(2). As explained, it is the sum of query evaluations using given variables. And I can imagine some scenarios where the proposed variable score may not behave as expected."
>
> R1: We are very sorry for our carelessness. What we actually used is the average of query evaluations using given variables (please see line 134 in the code of "Node.py" in the supplemental files), instead of the sum presented as Eq. (2) in the original paper. We now have revised it. That is, the variable score is
> $$
> \begin{align*}
> \boldsymbol{s}=\left(\sum_{(\mathbb M, \mathcal D)\in \mathbb D} g(\mathbb M) \odot \left(\sum_{(\boldsymbol{x}^i,y^i) \in \mathcal D} y^i \cdot \boldsymbol 1_D\right)\right) \big / \left( \sum_{(\mathbb M, \mathcal D)\in \mathbb D} |\mathcal D| \cdot g(\mathbb M) \right),
> \end{align*}
> $$
> where $/$ is the element-wise division, each dimension of $\sum_{(\mathbb M, \mathcal D)\in \mathbb D} g(\mathbb M) \odot (\sum_{(\boldsymbol{x}^i,y^i) \in \mathcal D} y^i \cdot \boldsymbol 1_D)$ is the sum of query evaluations using each variable and each dimension of $\sum_{(\mathbb M, \mathcal D)\in \mathbb D} |\mathcal D| \cdot g(\mathbb M)$ is the number of queries using each variable. With the average operation, the mentioned problems will not happen. Thank you very much for pointing out this issue.
>
> Q2: "It seems that $r$ in Theorem 4.2 appearing in $\beta_t$ is another hyperparameter ... Can any comment be added?"
>
> R2: $r$ is the upper bound on each variable, as we assumed that the domain $\mathcal X\subset [0, r]^D$ in line 223 in the original version (i.e, line 227 in the revised version). To make it more clear, we have revised to add the explanation in Theorem 4.2.
>
> Suggestion 1: "Even though the provided regret analysis is, in a sense, novel, ... this bound is quite loose. … However, this may be a good intermediate step to the bound which tells us the effect of variable selection methods in making the cumulative regret sublinear. It would be better if this point can be addressed."
>
> R1: Thank you for your suggestion. We fully agree that it is very interesting to perform a more specific theoretical analysis which can guarantee sublinear regret, as also suggested by Reviewers EcyS and do8w. As you suggested, the current general regret bound can be an intermediate step for this purpose. A possible way is to analyze the bound on the number of unselected important variables at each iteration of MCTS-VS by assuming some properties of the problem to be optimized, specifying the condition on the hyperparameters of MCTS-VS and using the UCB-based node selection of MCTS; then combine it with the general regret bound derived in the paper. To be honest, we tried to perform this specific analysis, but cannot finish it currently. In the last paragraph of Conclusion, we have revised to clearly indicate this limitation and leave it as our future work.

---

> > ### Comment · Reviewer_SbWH · 2022-08-08
> > **Thanks for the rebuttal**
> >
> > The authors address my concerns well. Even though some concerns about the empirical evaluation were raised by other reviewers and AC, still I am inclined to the acceptance of this paper. It seems that other reviewers also have concerns about the regret analysis. I think the general algorithmic idea outweighs the regret analysis. However, it seems appropriate to discuss the limitation of this proof appropriately in a revised version.

---

> > > ### Author Response · Authors · 2022-08-08
> > > **Response to Reviewer SbWH 2**
> > >
> > > Thank you. We will discuss the limitation of the proof appropriately in the final version.

---

> ### Author Response · Authors · 2022-08-02
> **Response to Reviewer SbWH (2/2)**
>
> Suggestion 2: "The method relies on the assumption of 'low effective dimensionality. … I admit that the MuJoCo experiments support that MCTS-VS may perform well for the function without low effective dimensionality. Still, it would be interesting to see the performance of MCTS-VS on some problems which are known not to have low effective dimensionality. A discussion on this can be addressed in a revised version."
>
> R2: Yes, our method relies on the assumption of low effective dimensionality, and might not work well if the percentage of valid variables is high. As also suggested by Reviewer xcd6, we have revised to add an experiment to examine the performance of MCTS-VS when the percentage of valid variables increases. We use the synthetic function Hartmann$6$\_$500$, and generate the variants with more valid variables by mixing multiple Hartmann$6$ functions. For example, Hartmann$6$\_$5$\_$500$ is generated by mixing five Hartmann$6$ functions as Hartmann$6(\boldsymbol x_{1:6}) +$ Hartmann$6(\boldsymbol x_{7:12})+\cdots+$ Hartmann$6(\boldsymbol x_{25:30})$, and appending $470$ unrelated dimensions, where $\boldsymbol x_{i: j}$ denotes the $i$-th to $j$-th variables. The other functions are generated alike. We have compared MCTS-VS-TuRBO with LA-MCTS-TuRBO and TuRBO on Hartmann$6$\_$500$, Hartmann$6$\_$5$\_$500$, Hartmann$6$\_$10$\_$500$, …, Hartmann$6$\_$30$\_$500$, and Hartmann$6$\_$83$\_$500$, which has the largest number (i.e., $6\times 83=498$) of valid variables. The results are shown in Figure 15 in Appendix E of the revised version. As expected, when the percentage of valid variables is low (e.g., in Hartmann$6$\_$500$, Hartmann$6$\_$5$\_$500$ and Hartmann$6$\_$10$\_$500$), MCTS-VS-TuRBO can be better than TuRBO; but as the percentage of valid variables increases, TuRBO becomes better, because a leaf node of MCTS can contain only a small fraction of valid variables. Thanks to your suggestion, we have also revised to emphasize the assumption of low effective dimensionality of our method in the last paragraph of Conclusion.
>
> For the good performance of MCTS-VS in the MuJoCo experiments, it may be because optimizing only a subset of variables is sufficient for achieving the goal, though each variable is valid. For example, the Walker2D robot consists of four main body parts: a torso, two thighs, two legs and two feet, where adjacent ones are connected by two hinges. The goal is to move forward by optimizing the hinges, each of which is valid. But even locking the hinges between legs and feet, the robot can still move forward by optimizing the other hinges. This is similar to that when the ankles are fixed, a person can still walk. We have revised to add some explanation in Appendix E of the revised version. Thank you very much.

---

### Official Review · Reviewer_EcyS · 2022-07-10

**Rating:** 6
**Confidence:** 2
**Soundness:** 3 good
**Presentation:** 2 fair
**Contribution:** 3 good

**Summary:**

The paper proposes a Monte-Carlo Tree Search (MCTS) approach for variable selection in Bayesian optimization (BO). A key challenge in BO is the optimization of the acquisition function to compute the next point to be evaluated, and this optimization procedure may suffer from the curse of dimensionality (i.e. require a number of evaluations of the acquisition function that grows quickly with the dimension).  The paper tackles this issue by applying MCTS to focus the acquisition on smaller subsets of variables, and decrease the computation cost of the optimization of the acquisition function.

**Questions:**

Suggestions:
* In Algorithm 1, several points are not clear, so that the reader needs to look up details in the main text. For instance, here are some points where I believe that more clarification is needed (either in Algorithm 1, or by putting clear references to their definitions in the main text):
	* The parameter $k$ is not mentioned in the list of parameters
	* Line 12: "update $n_{\mathrm{bad}}$
	* Line 20: "update the best $k$ points sampled so far"

* Including a definition of the regret in section 4 would improve the presentation
* Figures 1 and 2 are too small. Maybe section 3 could be shortened by making the presentation of Algorithm 1 clearer, which might create more space for the figures.
* In some parts of the paper (e.g line 59, section 2.2) when talking about the computational complexity of BO, it is not clear whether you refer to the complexity of optimising the acquisition function (which depends on the dimension and is tackled in the scope of the paper, if I understood correctly), and the complexity of keeping/updating a GP model (which depends mostly on the number of samples). Clearly making this distinction would be helpful for understanding the contributions of the paper.


**Limitations:**

Regarding the limitations, I suggest the authors add more details about how to choose the hyperparameters of the algorithm. Although they provide some sensitivity analyses in the appendix, one of the limitations of the proposed approach is the amount of hyperparameters required and their impact on the runtime (in addition to their impact on the performance of the optimization). A more in-depth theoretical analysis could help us understand the impact of the hyperparameters and how to choose them.

**Strengths And Weaknesses:**

Strengths:
* The paper provides a novel approach for variable selection in Bayesian optimisation
* The paper provides an theoretical analysis of the regret of algorithms that, at each time t, select only a subset of variables to optimise the acquisition function.
* The approach is validated empirically, showing that in a set of benchmarks the proposed algorithm achieves comparable or better results than the baselines, and verifies that it indeed brings a speed-up with respect to vanilla BO, although slower (as expected) than Dropout BO that does not perform adaptive variable selection.

Weaknesses:
* The regret bound seems to be a simple generalisation of the analysis of Dropout-BO (Li et al. 2017) and it is quite generic: it depends on which variables are selected at each time t, so it does not give many insights about the behaviour of the proposed algorithm, notably with respect to the choice of hyperparameters. I believe that an important advantage of presenting theoretical analyses for such kinds of algorithms is to help us understand the effects of hyperparameters on its performance, which is not possible with the generic analysis presented in the paper.
* As I detail in suggestions below, the presentation of the paper could be improved.

---

> ### Author Response · Authors · 2022-08-02
> **Response to Reviewer EcyS**
>
> Thank you for your positive review. We are very grateful to you for carefully reviewing our paper and providing constructive comments and suggestions.
>
> We fully agree that it is very interesting to perform a specific theoretical analysis about MCTS-VS, as also suggested by Reviewers SbWH and do8w. As Reviewer SbWH suggested, the current general regret bound can be an intermediate step for this purpose. A possible way is to analyze the bound on the number of unselected important variables at each iteration of MCTS-VS by assuming some properties of the problem to be optimized, specifying the condition on the hyperparameters of MCTS-VS and using the UCB-based node selection of MCTS; then combine it with the general regret bound derived in the paper. Such a specific theoretical result would be more helpful for understanding the influence of hyperparameters on the performance of MCTS-VS, as you indicated. To be honest, we tried to perform this specific analysis, but cannot finish it currently. In the last paragraph of Conclusion, we have revised to clearly indicate this limitation and leave it as our future work.
>
> For the selection of hyperparameters, our experiments of sensitivity analysis showed that the performance of MCTS-VS is not sensitive to most hyper-parameters. Thus, the default hyperparameter settings recommended in Appendix can be used in practice. To make further improvement, the hyperparameters can be fine-tuned carefully according to the problem property. For example, if the valid dimensionality is a little higher, the value of $N_{split}$ (the threshold for splitting a node) can be increased, which will allow to optimize more variables in a leaf node. But as you indicated, the amount of hyperparameters required by MCTS-VS may be a limitation, which we have revised to clearly indicate in the last paragraph of Conclusion.
>
> Thanks to your suggestion, we have also revised to add some discussion about the influence of hyperparameters on the runtime of MCTS-VS. The parameter $N_{split}$, i.e., the threshold for splitting a node, has a direct impact on the runtime, because it controls the least number of variables to be optimized in a leaf node. Other parameters may affect the depth of the tree and thus the runtime. For example, if the parameter $N_{bad}$ (the threshold of re-initializing the tree) is set to a very small value, MCTS-VS re-builds the tree frequently and the depth of the tree is small. The node with a small depth has more variables, leading to more runtime to optimize. For more details, please see the last paragraph in Appendix D of the revised version.
>
> We have revised the presentation issues according to your suggestions. But it is difficult to enlarge Figures 1 and 2 in the main paper due to space limitation, and we have revised to add their enlarged versions in Appendix F, i.e., Figures 17 and 18. Thank you very much.

---

> > ### Comment · Reviewer_EcyS · 2022-08-09
> > **Response**
> >
> > Thank you very much for you detailed response, which address all my main concerns!

---

### Official Review · Reviewer_xcd6 · 2022-07-11

**Rating:** 7
**Confidence:** 3
**Soundness:** 3 good
**Presentation:** 4 excellent
**Contribution:** 4 excellent

**Summary:**

The paper investigates a Monte Carlo tree search (MCTS) strategy of finding a subset of the available variables in a black-box optimization problem to perform Bayesian optimization (BayesOpt) on.
The algorithm first maintains a vector of scores that quantifies how important each variable is according to observed data.
It then selects the leaf node for further inspection using UCB, where each leaf node represents a subset of variables to use for optimization.
After this selection, the algorithm uses a separate BayesOpt method (e.g., TuRBO) to perform the actual optimization and bifurcates the leaf node appropriately.
The authors also present theoretical results for the bound on the cumulative regret of the algorithm, showing a connection to that of Dropout, which is a simple competitor.
Finally, experiment results show that their algorithm is competitive against a wide range of baselines in the literature, including a MCTS that limits the range of each variable.

**Questions:**

A couple of questions I have are about the failure modes of MCTS-VS and how it performs against LA-MCTS.
The authors mentioned that the approach might not work well if the percentage of valid variables is high (for example, in Levy experiments).
This is reasonable, and I would be interested to see a synthetic experiment where this percentage could be set to observe the performance of the algorithm as a function of this ratio.

Further, this doesn't mean that the proposed algorithm is not well-motivated, but simply that it isn't appropriate for some functions.
However, could there be ways to automatically flag this during optimization (by inspecting how the tree is being updated and the scores of the leaf nodes maybe)?
I imagine being able to automatically output a message that variable selection might not be useful on a specific function would be a nice feature.

I find it interesting that there are many cases in which MCTS-VS-BO performs better than the TuRBO variant, given how TuRBO has been shown to do well on a wide range of high-dimensional tasks.
Perhaps MCTS-VS is successfully picking out a small set of variables, and doing vanilla BO on this low-dimensional space is more appropriate than using TuRBO.
Did the authors try using LA-MCTS-BO in their experiments?

As a small note, I was hoping more insight could be extracted from the theoretical results.
Are there interpretations about when we could expect the proposed algorithm to have a lower regret bound than Dropout, maybe at some specific settings of $B_T$ or $\alpha_i^*$?

**Limitations:**

Yes, the paper mentions its limitations in the Results section and in the appendix.

**Strengths And Weaknesses:**

This is a well-written paper.
The exposition is clear, and the various components of the algorithm are well-motivated.
The problem of high-dimensional BayesOpt is well-motivated: (1) the curse of dimensionality makes requires exponentially more samples to identify the global optimum, and (2) training and prediction with a Gaussian process (GP) scales poorly.
The approach proposed here is promising because it allows narrowing down a small set of tunable parameters that are important (with respect to optimization), making it amenable to a budgeted search with a GP.
It is clear from their experiment results that their algorithm performs well across various tasks compared to other benchmarks in the literature.

I don't have any major complaints.
The proposed approach seems to be a natural complement to the cited LA-MCTS, which limits the range of the variables, as opposed to finding a subset of them.
Most of my comments and questions are related to the comparison between these two algorithms and are included in the Questions section.

Overall, the submission addresses an important problem and the proposed solution seems to work well on the inspected settings, and I lean towards acceptance.

---

> ### Author Response · Authors · 2022-08-02
> **Response to Reviewer xcd6 (1/2)**
>
> Thank you for your positive review. We are very grateful to you for carefully reviewing our paper and providing constructive comments and suggestions. Below please find our response.
>
> Q1: "A couple of questions I have are about the failure modes of MCTS-VS and how it performs against LA-MCTS. The authors mentioned that the approach might not work well if the percentage of valid variables is high (for example, in Levy experiments). This is reasonable, and I would be interested to see a synthetic experiment where this percentage could be set to observe the performance of the algorithm as a function of this ratio."
>
> R1: Thanks to your suggestion, we have revised to add an experiment to examine the performance of MCTS-VS when the percentage of valid variables increases. We use the synthetic function Hartmann$6$\_$500$, and generate the variants with more valid variables by mixing multiple Hartmann$6$ functions. For example, Hartmann$6$\_$5$\_$500$ is generated by mixing five Hartmann$6$ functions as Hartmann$6(\boldsymbol x_{1:6})+$ Hartmann$6(\boldsymbol x_{7:12})+\cdots+$ Hartmann$6(\boldsymbol x_{25:30})$, and appending $470$ unrelated dimensions, where $\boldsymbol x_{i: j}$ denotes the $i$-th to $j$-th variables. The other functions are generated alike. We have compared MCTS-VS-TuRBO with LA-MCTS-TuRBO and TuRBO on Hartmann$6$\_$500$, Hartmann$6$\_$5$\_$500$, Hartmann$6$\_$10$\_$500$, …, Hartmann$6$\_$30$\_$500$, and Hartmann$6$\_$83$\_$500$, which has the largest number (i.e., $6\times 83=498$) of valid variables. The results are shown in Figure 15 in Appendix E of the revised version. It can be observed that LA-MCTS-TuRBO performs the worst. As expected, when the percentage of valid variables is low (e.g., in Hartmann$6$\_$500$, Hartmann$6$\_$5$\_$500$ and Hartmann$6$\_$10$\_$500$), MCTS-VS-TuRBO can be better than TuRBO; but as the percentage of valid variables increases, TuRBO becomes better because a leaf node of MCTS can contain only a small fraction of valid variables.
>
> Q2: "Further, this doesn't mean that the proposed algorithm is not well-motivated, but simply that it isn't appropriate for some functions. However, could there be ways to automatically flag this during optimization. ... I imagine being able to automatically output a message that variable selection might not be useful on a specific function would be a nice feature."
>
> R2: Thanks for your suggestion. This is a very good idea that MCTS-VS automatically outputs a message when it is not useful. A natural and straightforward way may be using the number of re-initializing the tree. By our implementation, if the number of visiting a right child node (regarded as containing unimportant variables) is larger than a threshold $N_{bad}$, the Monte Carlo tree is regarded as built improperly and will be re-initialized. Thus, if the tree is re-initialized frequently, it implies that MCTS cannot do a good variable selection for the current problem, and then a corresponding message can be output. We have revised to mention this at the end of Section 3.
>
> Q3: "I find it interesting that there are many cases in which MCTS-VS-BO performs better than the TuRBO variant, ... Perhaps MCTS-VS is successfully picking out a small set of variables, and doing vanilla BO on this low-dimensional space is more appropriate than using TuRBO. Did the authors try using LA-MCTS-BO in their experiments?"
>
> R3: For LA-MCTS, it is recommended to use TuRBO as the default optimizer [Wang et al., 2020]. Thus, we only used LA-MCTS-TuRBO for comparison in the experiments. Thanks to your suggestion, we have run LA-MCTS-BO in the experiments. But due to time limit, we have only got the result on Hartmann$6$\_$300$. The superior performance of LA-MCTS-TuRBO over LA-MCTS-BO is consistent with previous observation [Wang et al., 2020].
>
> [1] L. Wang, R. Fonseca, and Y. Tian. Learning search space partition for black-box optimization using Monte Carlo tree search. In Advances in Neural Information Processing Systems 33 (NeurIPS’20), pages 19511–19522, Vancouver, Canada, 2020.

---

> > ### Comment · Reviewer_xcd6 · 2022-08-04
> > **Response to the authors**
> >
> > Thank you for the detailed responses, especially on the point about the experiment with variable percentage of valid variables.

---

> ### Author Response · Authors · 2022-08-02
> **Response to Reviewer xcd6 (2/2)**
>
> Q4: “As a small note, ... Are there interpretations about when we could expect the proposed algorithm to have a lower regret bound than Dropout ...?
>
> R4: Thanks for your suggestion. Dropout selects $d$ variables randomly at each iteration. According to Theorem 4.2, one direct condition for lower regret bound of MCTS-VS can be that given large enough $T$, MCTS-VS selects at least $d$ variables (i.e., $|\mathbb M_t| \geq d$) at each iteration and the unselected variables have smaller values of $\alpha^*_i$ (i.e., are less important). We also want to admit that the theoretical analysis can be improved. The current analysis is for general variable selection, while a specific regret analysis for MCTS-VS will be very interesting, as suggested by Reviewers EcyS, SbWH and do8w. To be honest, we tried to perform the specific analysis, but cannot finish it currently. In the last paragraph of Conclusion, we have revised to clearly indicate this limitation and leave it as our future work. Thank you very much.

---

### Official Review · Reviewer_wG4G · 2022-07-14

**Rating:** 7
**Confidence:** 3
**Soundness:** 3 good
**Presentation:** 3 good
**Contribution:** 3 good

**Summary:**

This paper proposes a novel variable selection algorithm based on MCTS and BO.
The proposed method, MCTS-VS, outperforms existing approaches in several benchmark problems with relatively large dimensions.


**Questions:**

Can you say that at least in some settings (e.g., some type of 100-1000 dimensional problem), this experimental comparison is sufficient?
Are other well-known variable-selection methods are not applicable or not suitable for these benchmarks?

---
Post rebuttal comments.

Thank you for the clarification.
I changed my score.

**Limitations:**

The authors explained the limitations related to the hyperparameters in the main material and the appendix.


**Strengths And Weaknesses:**

A novel variable selection algorithm outperforms existing work in benchmarks with relatively large dimensions.

The algorithm and the experiments are well described, which should be useful for others in both theory and practice.

Weaknesses:
There are numerous variable selection methods, some of which are useful for high-dimensional problems (such as LASSO-based approaches).
However, this paper only compares against Dropout based methods.
It is not surprising to see MCTS outperforming random selection.
I cannot say that I am fully convinced of the superiority of the proposed method because the comparison is limited.

---

> ### Author Response · Authors · 2022-08-02
> **Response to Reviewer wG4G**
>
> Thank you for your positive review. We are very grateful to you for carefully reviewing our paper and providing constructive comments and suggestions. Below please find our response.
>
> Suggestion: "Compare with other variable selection methods"
>
> R: Yes, other variable selection methods can be directly employed. But to the best of our knowledge, only Dropout (i.e., random selection) has been used for high-dimensional BO. This may be because many of existing variable selection methods (e.g., LASSO, forward regression and OMP) usually require a large number of samples to fit the linear regression model well, while in BO scenarios, only a limited number of samples can be evaluated. Thanks to your suggestion, we have revised to add a comparison with LASSO-based variable selection, named LASSO-VS. We use the synthetic function Hartmann$6$\_$300$. The results are shown in Figure 16 in Appendix E of the revised version. When equipped with either BO or TuRBO, the proposed MCTS-VS always performs the best. When equipped with BO, LASSO-VS can even be worse than Dropout.

---

### Author Response · Authors · 2022-08-02
**General response to reviewers**

We are very grateful to the reviewers for carefully reviewing our paper and providing constructive comments and suggestions. We are also very glad that all the reviewers give positive reviews. We have revised the paper carefully according to the comments and suggestions. Revisions have been colored red in the revised paper for the sake of clarity. Our response to individual reviewers can be found in the personal replies, but we also would like to make a brief summary of revisions for your convenience.

We have revised to add some experiments, to examine the performance of the proposed MCTS-VS method when the percentage of valid variables increases, to compare MCTS-VS with the LASSO-based variable selection method, and to examine the effectiveness of a new "fill-in" strategy, i.e., the average best-$k$ strategy. We have also revised to add more explanation about the good performance of MCTS-VS in the MuJoCo experiments as well as the influence of the hyperparameters on the runtime. We have revised to add one paragraph in Conclusion to clearly indicate the limitation of our work.

For the other minor comments, we have also revised carefully. We want to thank the reviewers again for providing constructive comments and suggestions, which have improved the paper.

---

### Comment · Area_Chair_aR8v · 2022-08-07
**Feature selection via sparsity**

The reviewers were generally positive about this paper, but this work lacks a comparison the most relevant SoTA algorithm for high-dimensional Bayesian optimization, the [SAASBO method of Erikson & Jankowiak, 2021](https://arxiv.org/pdf/2103.00349.pdf) which performs feature selection via a sparsity inducing prior.  The authors do briefly cite this work, but say that the prior "restrains over-exploration". However, the framing of the SAASBO paper is all about selecting important features and penalizing unimportant ones.  SAASBO has been shown to outperform TuRBO—the most competitive baseline algorithm in this work, and the code is publicly available.  A comparison with this algorithm is crucial for developing an understanding of the benefits and tradeoffs with each of these algorithms.

---

> ### Author Response · Authors · 2022-08-08
> **Response to Area Chair aR8v**
>
> Dear Area Chair aR8v,
>
> Thank you for your comment. SAASBO employs sparsity-inducing function prior and No-Turn-U-Sampler (NUTS) to perform variable selection, making the unimportant variables not selected to optimize and thus restraining the over-exploration. We are sorry that we did not present SAASBO precisely. We have revised to make it clear in the uploaded new version.
>
> Thanks to your suggestion, we have revised to add a comparison between MCTS-VS-BO and SAASBO on Hartmann$6$\_$100$ and Hartmann$6$\_$500$. For a fair comparison, the acquisition function optimization of SAASBO is the same as that (i.e., randomly generate numerous points and select some ones with the maximal expected improvements, which is similar to the implementation in [LA-MCTS](https://github.com/facebookresearch/LaMCTS), [TuRBO](https://github.com/uber-research/TuRBO) and [HeSBO](https://github.com/aminnayebi/HesBO)) of the methods compared in our original experiments, and its other hyperparameters use the default values in SAASBO. Due to time limitation, we only run 100 and 150 evaluations for Hartmann$6$\_$100$ and Hartmann$6$\_$500$, respectively. The average results over the random seeds (2021--2023) are shown in Figure 19 in Appendix of the uploaded new version. It can be observed that SAASBO has similar performance to MCTS-VS-BO.
>
>
> However, when considering the practical running overheads, SAASBO consumes much more time than MCTS-VS-BO. For Hartmann$6$\_$100$, the wall clock time of 100 iterations of MCTS-VS-BO and SAASBO averaged over the three random seeds (2021--2023) are $2.327$s and $687.437$s, respectively. For Hartmann$6$\_$500$, the time of 150 iterations of MCTS-VS-BO and SAASBO are $7.827$s and $25532.613$s, respectively. The experiments of comparing wall clock time are conducted on Intel(R) Core(TM) i7-10700 CPU @ 2.90GHz and use single thread. This observation is consistent with the previous one. For example, Table 1 in [Eriksson and Jankowiak, 2021] shows that SAASBO costs much more time than other methods, e.g., the runtime of SAASBO (default) and SAASBO (128-128-8) per iteration are $26.51$s and $19.21$s, respectively, while that of TuRBO is $1.52$s. SAASBO is very time-consuming due to the high computational cost $\mathcal O(t^3D)$ of obtaining posterior samples by NUTS for inference, where $t$ is the number of data points and $D$ is the dimension of the optimization problem. Thus, when using the wall clock time instead of the number of evaluations as the $x$-axis, the advantage of MCTS-VS-BO over SAASBO is very clear, as shown in Figure 20 in Appendix of the uploaded new version.
>
> We also would like to emphasize the idea of the proposed variable selection method MCTS-VS. MCTS-VS uses the Monte Carlo tree to iteratively partition the variables into important and unimportant ones, and optimizes only those selected important variables via a black-box optimization algorithm. This process does not utilize any information from the black-box optimization algorithm (note that SAASBO uses a GP prior to perform variable selection). Thus, MCTS-VS is a **general** framework to do variable selection and can be equipped with **any** optimization algorithm. For example, we have equipped MCTS-VS with vanilla BO and TuRBO in the paper, showing that MCTS-VS can bring improvement.
>
> In fact, SAASBO may be also used under the MCTS-VS framework to further improve the performance, as mentioned in line 93 in the revised paper. MCTS-VS equipped with SAASBO is briefly called MCTS-VS-SAASBO. We have revised to run MCTS-VS-SAASBO on Hartmann$6$\_$500$. The results are shown in the right sub-figure of Figure 19. The performance of SAASBO and MCTS-VS-SAASBO is similar. But when considering the runtime, the time of 150 iterations of MCTS-VS-SAASBO is $4779.109$s, while the time of SAASBO is $25532.613$s. That is, MCTS-VS-SAASBO can achieve about $5$ times acceleration. The curves of using the wall clock time as the $x$-axis in the right sub-figure of Figure 20 clealy show the advantage of MCTS-VS-SAASBO over SAASBO. MCTS-VS-SAASBO selects the variables containing important ones by MCTS and then uses SAASBO to optimize the selected variables, which reduces the dimension and thus costs much less time than using SAASBO directly.
>
> The combination of MCTS-VS and SAASBO is similar to a hierarchical variable selection method, i.e., MCTS-VS first performs an efficient but rough variable selection to select some variables, and then SAASBO performs a time-consuming but precise variable selection under the relative low-dimensional space, to further select the important variables. It may be a potential solution for BO to handle extremely high-dimensional optimization problems, which is difficult to select important variables directly.
>
> We will add the comparison with SAASBO in the final version, and also discuss the benefit of combining MCTS-VS with SAASBO. Thank you.

---

### Meta-Review · Area_Chair_aR8v · 2022-08-28

**Recommendation:** Accept
**Confidence:** Less certain

**Metareview:**

This work proposes a variable selection technique based on MCTS that can be integrated with various acquisition functions.  The authors show that this can significantly speed up the optimization of acquisition functions in Bayesian optimization, and in some cases, improve the optimization performance as well.  The reviewers were quite positive about the paper, but few expressed high confidence in their rating.

There were some reservations that the work is not adequately benchmarked against recent algorithms for high-dimensional BayesOpt that include aspects of feature selection (e.g., SAASBO). During the rebuttal, the authors showed that MCTS-VS methods could successfully be used in combination with SAASBO, and that this can improve wall time performance.  I hope that the authors can explore this in more detail in the final version.

The empirical results could be presented in a more rigorous and transparent fashion.  The authors provide compelling evidence that MCTS-VS greatly speeds up "wall clock time". Table 1 does a good enough of a job of communicating the runtime advantages (although it is not clear if this comes from model fitting or AF optimization time).  Such speedups can be particularly helpful in higher throughput scenarios when thousands of iterations occur, or one wishes to use more compute-intensive models, such as the SAAS model.

The goal of Bayesian optimization is to efficiently perform global optimization of expensive-to-evaluate functions.  The performance with MCTS-VS with respect to this goal is a little less consistent, but still promising.  This can be seen in Fig 11 from the rebuttal, where there is no significant difference between most HDBO methods in terms of BayesOpt performance (Fig 3, in this respect, is a somewhat out of place and might mislead readers; I would recommend eliminating such plots, since walltime vs performance plots are really only meaningful for cases like multi-fidelity BO or BO with early stopping, where wall-time refers to the expensive-to-evaluate functions).

Finally, real-world problems likely do not contain many parameters that are truly irrelevant in the same way as the synthetic problems do.  Including more examples of real-world problems in the main text can highlight the practical benefits of this algorithm.  I would recommend using real estate from e.g., the Levy problem, to explore other high-dimensional problems, such as feature selection type problems like the SVM benchmark in the SAASBO paper.

**Award:**

No

---

### Decision · Program_Chairs · 2022-09-14

Accept